# An Algorithm of Image Encryption Using Logistic and Two-Dimensional Chaotic Economic Maps

**DOI:** 10.3390/e21010044

**Published:** 2019-01-09

**Authors:** Sameh S. Askar, Abdel A. Karawia, Abdulrahman Al-Khedhairi, Fatemah S. Al-Ammar

**Affiliations:** 1Department of Statistics and Operations Researches, College of Science, King Saud University, P.O. Box 2455, Riyadh 11451, Saudi Arabia; 2Department of Mathematics, Faculty of Science, Mansoura University, Mansoura 35516, Egypt; 3Computer Science Unit, Deanship of Educational Services, Qassim University, P.O. Box 6595, Buraidah 51452, Saudi Arabia

**Keywords:** logistic map, chaotic economic map, image encryption, image decryption, security analysis, 68P25, 68U10, 94A60

## Abstract

In the literature, there are many image encryption algorithms that have been constructed based on different chaotic maps. However, those algorithms do well in the cryptographic process, but still, some developments need to be made in order to enhance the security level supported by them. This paper introduces a new cryptographic algorithm that depends on a logistic and two-dimensional chaotic economic map. The robustness of the introduced algorithm is shown by implementing it on several types of images. The implementation of the algorithm and its security are partially analyzed using some statistical analyses such as sensitivity to the key space, pixels correlation, the entropy process, and contrast analysis. The results given in this paper and the comparisons performed have led us to decide that the introduced algorithm is characterized by a large space of key security, sensitivity to the secret key, few coefficients of correlation, a high contrast, and accepted information of entropy. In addition, the results obtained in experiments show that our proposed algorithm resists statistical, differential, brute-force, and noise attacks.

## 1. Introduction

Converting secret information such as images from a decipherable form to an undecipherable one has been recently investigated to fulfill the demand of a secure route of image transmission through different communications channels such as Internet and wireless networks. Encryption schemes have been introduced based on chaotic models to handle that issue and to provide such secure routes [1,2]. The Data Encryption Standard (DES) algorithm is a traditional scheme and has been used for encrypting images [3]. It has some drawbacks, among which is that the algorithm’s efficiency is low when dealing with large images [4,5]. The theory of chaos is an important theory that possesses some complex features such as sensitivity to the initial conditions, unpredictability, and bifurcation. It has been recently adopted by researchers in the process of encryption [1,2]. Chaos has presented some popular models, which include some complex properties that help provide a robust encryption scheme. Such models include the logistic map, Lorenz attractor, the Hénon map, and some chaotic economic systems [6]. Such complex properties of chaotic maps may be reflected in an analogy to certain properties of the cryptographic process of ideal ciphers like diffusion, confusion, balance, and avalanche.

The cipher schemes proposed based on chaotic characteristics have shown a robust and efficient way to tackle image broadcast very quickly with highly secured routes via different types of telecommunications. Since Matthews [7] published his first algorithm of encryption in 1989, which depended on some chaotic characteristics, a number of chaos-based image encryption schemes with have been developed in literature. For instance, in [8], the authors created a cipher image by dividing it into some blocks and used some kind of permutations applied to each block using the logistic map. A two-dimensional discrete chaotic system has been introduced in [9]. It used the Chebyshev chaotic sequences to scramble each pixel in the novel image. Increasing the key space is another aspect that makes the algorithm used for encryption very difficult to break, and this has been performed in [10]. A multi-chaotic system has been adopted in the process of encryption in [10]. In [11], the Rossler system has been included in an encryption algorithm and has been used to change the image’s pixel values and position in order to make the ciphered images have high uncertainty. A logistic chaotic function has been used in [12] with the one-time pads in order to increase the encryption space, and hence, robust cipher images can be obtained. The shuffling approach has been introduced in [13], presenting a good scheme of encryption based on the cat map. It is a one-dimensional chaotic map with some advantages such as simplicity and high efficiency [14]. However, it has some weaknesses that may make it possible for the encryption scheme to be easily broken. This weakness includes a small key space and a weak security [15,16].

It is important here to highlight some of the recent and state-of-the-art algorithms in cryptography that will be used in comparison with our proposed algorithm. For example, in [17], Sivakumar et al. introduced a new image encryption algorithm based on pixel position permutation and a random key stream. In [18], the authors combined DNA sequences with a chaotic map to introduce an image algorithm of encryption. Based on discrete wavelet transform and multi-chaos, an image algorithm was proposed in [19]. The Arnold transformation has been used to encrypt images with high security in [20]. The idea in [20] depended on splitting the image randomly into several parts, and then, each part was coded by the Arnold transformation to increase the security level. Adopting the classical confusion-diffusion structure, an image encryption algorithm has been introduced by Hua et al. based on a 2D logistic-sine-coupling map [21]. In [22], a simple piecewise linear chaotic map has been utilized to construct a bit-level image encryption algorithm. Recently, a robust image encryption/decryption algorithm via 1D chaotic economic map was proposed in [1]. The advantage of that algorithm is that the periodic recovery of the plain image is removable.

Some of the disadvantages existing in those algorithms have led us to propose a new 2D chaotic algorithm for image encryption. The algorithm published in [1] is constructed based on a one-dimensional chaotic map, but this one depends on a two-dimensional chaotic system. The number of parameters in the current algorithm is more than that in [1], and therefore, the security level is higher than the one in [1]. The current paper is organized as follows. In Section 2, a brief introduction about the 2D chaotic map used in the algorithm is presented. In Section 3, the algorithm’s steps are introduced in detail. In Section 4, we discuss our obtained results due to applying the algorithm on some popular images and make some comparisons with the results obtained by existing algorithms in the literature. In Section 5, we give some conclusions.

## 2. The Two-Dimensional Map

Recently, the adoption of chaotic dynamical systems to encrypt images has drawn researchers’ attention. Such systems possess many parameters that have influences on the systems’ behaviors and are suitable in the encryption scheme. Those parameters have some important usage, as they are used as security keys in the process of encryption. Small changes in the value of the system’s parameters may make the system enter a region where chaos arises. The chaotic behavior of such systems is an important aspect when cryptographic algorithms are constructed. In this work, we adopt the chaotic economic map introduced in [23] that takes the following form:(1)$$\left.\begin{array}{cc} x_{1,n+1} & =x_{1,n}+kx_{1,n}^{2}\left(a-c-b\frac{x_{1,n}}{Q_{n}}-b\log\left(Q_{n}\right)\right),\\ x_{2,n+1} & =x_{2,n}+kx_{2,n}^{2}\left(a-c-b\frac{x_{2,n}}{Q_{n}}-b\log\left(Q_{n}\right)\right), \end{array}\right\}$$ where:$$Q_{n}=x_{1,n}+x_{2,n},\;n=0,1,2,\ldots$$ Equation (Equation 1) shows the chaotic behavior of a map with six distinct parameters. These parameters are important in the economy. First, *a* stands for the market size and should always be positive, while the parameter b>0 represents the change of price within the market. Economically, the condition a>b and a>c should be always satisfied. The parameter c≥0 is called the cost of producing one unit of a good, and k>0 is known as the speed adjustment. The chaotic behavior of the economic map (1) is detected at the parameter values: $x_{1,0}=0.11,x_{2,0}=0.12,a=2,b=0.5$, and c=0.5. Figure 1a shows the bifurcation diagram of Equation (Equation 1) with respect to the parameter *k* for these parameters. Numerical simulation has shown that when the system (1) approaches the value k=0.6490, a two-cycle period appears, and so on, for other different cycle periods until the system becomes chaotic. This is clear in the Lyapunov exponents plotted in Figure 1c at the same values of the parameters. Lyapunov exponents tell us the rate of divergence of nearby trajectories, a key component of chaotic dynamics. Figure 1b shows the time series for both variables $x_{1,n}$ and $x_{2,n}$. Finally, Figure 1d shows the complex chaotic behavior of the system (1).

## 3. The Algorithm

It is known that any original image has strong correlations among its pixels. In the encryption process, it is important to scatter the pixels to break their correlations. The well-known approach to do that is the shuffling process, which is applied to each array of the original image. Suppose that the original image has the dimension M×N and $P={\left(p_{i,j}\right)}_{1\leq i\leq M,1\leq j\leq N}$ is the array of its pixel positions. To produce the shuffling array, ${\mathrm{Sh}}_{P}^{rc}$, we continue as follows:(Shuffling of rows) In this step, we recall the logistic map $x_{n+1}={\mathrm{rx}}_{n}\left(1-x_{n}\right)$. The logistic map is used to generate a set of random values in the interval [1,M], say $i_{1},i_{2},\ldots ,i_{M},\;i_{k}\neq i_{l}\;\forall \;k\neq l$. The rows of the P array are changed according to those random values, and hence, a new array ${\mathrm{Sh}}_{P}^{r}$ is obtained:
$${\mathrm{Sh}}_{P}^{r}=\left(\begin{array}{ccccc} p_{i_{1},1} & p_{i_{1},2} & \ldots & \ldots & p_{i_{1},N}\\ p_{i_{2},1} & p_{i_{2},2} & \ldots & \ldots & p_{i_{2},N}\\ \vdots & \vdots & \ldots & \ldots & \vdots \\ \vdots & \vdots & \ldots & \ldots & \vdots \\ p_{i_{M},1} & p_{i_{M},2} & \ldots & \ldots & p_{i_{M},N} \end{array}\right)$$(Shuffling of columns) Using a similar technique as in (1), we generate a new set of random values in the interval [1,N], say $j_{1},j_{2},\ldots ,j_{N},\;j_{k}\neq j_{l}\;\forall \;k\neq l$. The columns of ${\mathrm{Sh}}_{P}^{r}$ take the following form:
$${\mathrm{Sh}}_{P}^{rc}=\left(\begin{array}{ccccc} p_{i_{1},j_{1}} & p_{i_{1},j_{2}} & \ldots & \ldots & p_{i_{1},j_{N}}\\ p_{i_{2},j_{1}} & p_{i_{2},j_{2}} & \ldots & \ldots & p_{i_{2},j_{N}}\\ \vdots & \vdots & \ldots & \ldots & \vdots \\ \vdots & \vdots & \ldots & \ldots & \vdots \\ p_{i_{M},j_{1}} & p_{i_{M},j_{2}} & \ldots & \ldots & p_{i_{M},j_{N}} \end{array}\right)$$

Now, the steps of our encryption algorithm are constructed as follows:**Step** **1:**Read the original image, then convert it to a gray image, say P.**Step** **2:**Perform row and column shuffling to generate the shuffling array, say ${\mathrm{Sh}}_{P}^{rc}$.**Step** **3:**Convert the pixel values of ${\mathrm{Sh}}_{P}^{rc}$ from decimal to binary, $S=\{S_{1},S_{2},\cdots ,S_{MN}\}$.**Step** **4:**Use the two-dimensional chaotic economic map (1) to generate MN values $x=\{x_{1},x_{2},\cdots ,x_{MN}\}$ as follows:
**(i)** For *i* = 2:n, Compute: $\;x_{1,i}=x_{1,(i-1)}+kx_{1,(i-1)}^{2}\left(a-c-b\frac{x_{1,(i-1)}}{Q_{i-1}}-b\log\left(Q_{i-1}\right)\right)$
$\;x_{2,i}=x_{2,(i-1)}+kx_{2,(i-1)}^{2}\left(a-c-b\frac{x_{2,(i-1)}}{Q_{i-1}}-b\log\left(Q_{i-1}\right)\right)$. End. **(ii)** Set $x_{1,0}=x_{1,n}$ and $x_{2,0}=x_{2,n}$ as initial values. **(iii)** Generate MN values $x_{1}=\left\{x_{11},x_{12},\cdots ,x_{1,MN}\right\}$ and $x_{2}=\left\{x_{21},x_{22},\cdots ,x_{2,MN}\right\}$ using the two-dimensional chaotic economic map (1).**Step** **5:**Do the following preprocessing for the generated values in Step 4:
$$x_{i}=floor\left(mod\left(x_{i}\times 10^{14}\right),256\right)$$**Step** **6:**Convert the preprocessing values in Step 5 from decimal to binary, $B_{1}=\left\{b_{11},b_{12},\cdots ,b_{1,MN}\right\}$ and $B_{2}=\left\{b_{21},b_{22},\cdots ,b_{2,MN}\right\}$ for $x_{1}$ and $x_{2}$, respectively.**Step** **7:**Perform the bit-wise XOR between the values of $B_{1}$ and I, $e_{1}=bitxor\left(B_{1},I\right)$, where I equals the value of the array P after reshaping it to be a vector of size 1×MN.**Step** **8:**Perform the bit-wise XOR between the values of $B_{2}$ and I, $e_{2}=bitxor\left(B_{2},I\right)$.**Step** **9:**Perform the bit-wise XOR between the values of $e_{1}$ and $e_{2}$, $e=bitxor(e_{1},e_{2})$.**Step** **10:**Convert the values of e from binary to decimal, say E.**Step** **11:**The cipher pixel set is denoted by $E=\{E_{1},E_{2},\cdots ,E_{MN}\}$.**Step** **12:**Reshape the set E to be an array of size M×N, say C as the cipher image.

In addition, the steps of our decryption algorithm are described as follows:**Step** **1:**Read the cipher image, C with size M×N.**Step** **2:**Reshape C to the cipher pixel set $E=\{E_{1},E_{2},\cdots ,E_{MN}\}$.**Step** **3:**Convert the values of E from decimal to binary, say e.**Step** **4:**Repeat Step4 in the encryption algorithm to generate MN values for the two vectors, $x_{1}=\left\{x_{11},x_{12},\cdots ,x_{1,MN}\right\}$ and $x_{2}=\left\{x_{21},x_{22},\cdots ,x_{2,MN}\right\}$.**Step** **5:**Do the following preprocessing for the generated values in Step 4:
$$x_{i}=floor\left(mod\left(x_{i}\times 10^{14}\right),256\right)$$**Step** **6:**Convert the preprocessing values in Step 5 from decimal to binary, $B_{1}=\left\{b_{11},b_{12},\cdots ,b_{1,MN}\right\}$ and $B_{2}=\left\{b_{21},b_{22},\cdots ,b_{2,MN}\right\}$ for $x_{1}$ and $x_{2}$, respectively.**Step** **7:**Perform the bit-wise XOR between the values of $B_{1}$ and e, $F_{1}=bitxor\left(B_{1},e\right)$.**Step** **8:**Perform the bit-wise XOR between the values of $B_{2}$ and e, $F_{2}=bitxor\left(B_{2},e\right)$**Step** **9:**Perform the bit-wise XOR between the values of $F_{1}$ and $F_{2}$, $z=bitxor(F_{1},F_{2})$**Step** **10:**Convert the values of z from binary to decimal, say Z.**Step** **11:**Reshape Z to be an array of size M×N, say Sh.**Step** **12:**Perform row and column shuffling to get the decryption image, say D.

### The Secret Key Generation

Assume that $P=(p_{i,j})$, i=1,2,...,M,j=1,2,...,N, is the plain image, where $p_{i,j}$ refers to the value of the pixel at the position (i,j) and (M,N) is the size of the plain image P. The secret key will be calculated via the key mixing proportion factor as follows:(2)$$K_{u}=\frac{1}{256}mod\left(\sum_{i=\frac{(u-1)M}{10}+1}^{\frac{uM}{10}}\sum_{j=1}^{N}p_{i,j},256\right)$$

Then, we change the initial condition $\beta_{0}$ according to the following formula:(3)$$\beta_{0}\leftarrow \frac{\left(\beta_{0}+K\right)}{2},$$ where $\beta_{0}=x_{10},r_{10},x_{20},r_{20},x_{1,0},x_{2,0},a,b,c,k$ and $K=K_{u},u=1,2,\cdots ,10$, respectively.

After that, take two initial values for the logistic map, $x_{10},x_{20}$, two parameters for the logistic map, $r_{10},r_{20}$, two initial values for the system, $x_{1,0},x_{2,0}$, and four system parameters, a,b,c,k.

**Remark** **1.**
*It is not difficult to note that $0<K_{u}<1$. Therefore, the value of $\beta_{0}$ on the right-hand side of Equation (Equation 3) must satisfy the conditions of Map (1).*


## 4. Experimental Analysis

In the current section, we apply the introduced algorithm on several types of images to test its robustness. Ten well-known images, lena (128 × 128, 256 × 256, 512 × 512), barbara (256 × 256), airplane (512 × 512), boat (512 × 512), house (256 × 256), baboon (512 × 512), moon surface (256 × 256), and resolution chart (256 × 256) are tested as plain images. The plain images and their corresponding histograms are displayed in the first and second columns in Figure 2.

The security keys of our encryption algorithm are chosen as follows. In the logistic map, r=3.9985, $x_{0}=0.02$ for row shuffling and r=3.9995, $x_{0}=0.3$ for column shuffling. In the two-dimensional chaotic economic map (1), $x_{1,0}=0.11,x_{2,0}=0.12,a=2,b=0.5,c=0.5,$ and k=0.87. Our experiments are carried out under MATLAB R2016a running on a laptop with the following features: Intel(R) Core(TM) i7-4700MQ 2.40 GHz, 12.0 GB memory, and 1.0 TB capacity.

## 5. Security Analysis

A good encryption algorithm possesses the following properties: (i) histograms of cipher images that are completely uniform; (ii) good information entropy; (iii) very low relationship between two adjoining pixels; (iv) resistance to statistical and differential attacks; (v) sensitivity to the secret keys; (vi) huge key space; (vii) high contrast; (viii) resisting noise attacks. Those aspects are tested for the proposed algorithm.

### 5.1. Statistical Analysis

Statistically, plain and cipher images must be different and therefore similarity must be hidden in order to avoid the attempts of attackers to break the algorithm.

#### 5.1.1. Analysis of the Histogram

The distribution of pixel values in the image is shown by a histogram analysis. The histogram of the cipher image should be uniformly distributed. Therefore, obtaining any useful statistical information will be very hard. The third and fourth columns of Figure 2 display cipher images and their corresponding histograms.

The histograms of the obtained encrypted images are approximately uniform, are different from those histograms of the plain images, and show that the proposed algorithm is secure enough for image encryption. The cipher images have histograms that have an approximately uniform distribution. The chi-square test [24] is used to check this uniformity using the following formula:(4)$$\chi^{2}=\sum_{i=1}^{k}\frac{{\left(O_{i}-e_{i}\right)}^{2}}{e_{i}},$$ where the gray level number is k=256 and $O_{i}$ and $e_{i}$ represent the observed and the expected occurrence frequencies of each gray level (0–255). Using a level of significance that α=0.05, the chi-square values of the cipher images are given in Table 1.

It is easy to see that $\chi^{2}\left(k-1,\alpha \right)=\chi^{2}\left(255,0.05\right)=293$. Therefore, our null hypothesis is accepted with this level of significance, and the cipher histograms have a uniform distribution. It is shown in Table 1 that the value of $\chi^{2}$ is less than the tabulated value $\chi^{2}\left(255,0.05\right)$.

#### 5.1.2. Entropy Process of Information

For a source with symbol *S*, the process of entropy is denoted by *H* and is calculated by the following formula [25].
(5)$$H\left(S\right)=-\sum_{i=0}^{N-1}p\left(s_{i}\right)log_{2}p\left(s_{i}\right)$$ where $p(s_{i})$ refers to the probability of symbol $s_{i}$ and the bits are used for entropy. The theoretical result of entropy H(S) is eight if $N=256=2^{8}$ gray values for an image with equal probability. The information entropy shows the distribution of gray values. The information entropy for the cipher images is exhibited in Table 2. The obtained values are near the theoretical value of eight. Recently, the authors in [26] proposed a statistical test for the block entropy and introduced qualitative and quantitative results. We have selected 100 non-overlapped blocks of size 16×16 randomly from each cipher image. We have calculated the information entropy for each block based on Equation (Equation 5), and the average entropy was recorded. From Table 2, all actual block entropy values are passed the theoretical block entropy values at α=0.01 and α=0.05. Therefore, we may conclude that the cipher images is random-like after applying our proposed encryption algorithm.

#### 5.1.3. Examination of Correlation

In this subsection, the vertical, horizontal, and diagonal correlations between two adjacent pixels in a cipher image are evaluated. This examination can be carried out by selecting randomly 5000 pairs of two neighboring pixels from the original and cipher images. The coefficient of correlation for each pair is obtained using [4,27]:(6)$$r_{xy}=\frac{cov(x,y)}{\sqrt{D(x)}\sqrt{D(y)}}$$ where:$$cov\left(x,y\right)=\frac{1}{N}\sum_{i=1}^{N}\left(x_{i}-E\left(x_{i}\right)\right)\left(y_{i}-E\left(y_{i}\right)\right),$$
$$D\left(x\right)=\frac{1}{N}\sum_{i=1}^{N}\left(x_{i}-E\left(x_{i}\right)\right),$$ and:$$E\left(x\right)=\frac{1}{N}\sum_{i=1}^{N}x_{i}.$$ where the gray values of any two neighboring pixels of an image are denoted by *x* and *y*. Figure 3 demonstrates the horizontal distribution of the relationship between two neighboring pixels in the original image and its corresponding ciphered image. Table 3 shows the relationship of two neighboring pixels in the original and cipher images. Figure 3 presents those correlations for the plain and the cipher one. The correlation coefficients of the cipher image for our proposed algorithm and other algorithms are shown in Table 4. The correlation coefficients of two adjacent pixels of the cipher image are close to zero.

### 5.2. Sensitivity Analysis

One of the important features of a good encryption algorithm is the sensitivity to the security key and the plain image. Therefore, any small change in the security key or the plain image tends to a completely different cipher image [28].

#### 5.2.1. Differential Attack

In order to get some insights about the influence of changing one pixel in the cipher image, some measures such as NPCR (Number of Pixels Change Rate) and UACI (Unified Average Changing Intensity) are used. They have the following formulae:(7)$$\mathrm{NPCR}=\frac{\sum_{i,j}D\left(i,j\right)}{W\times H}\times 100\%,$$
(8)$$\mathrm{UACI}=\frac{1}{W\times H}\left[\sum_{i,j}\frac{|I_{1}\left(i,j\right)-I_{2}\left(i,j\right)|}{255}\right]\times 100\%.$$ where:(9)$$D\left(i,j\right)=\left\{\begin{array}{cc} 1 & \;\mathrm{if}\;I_{1}\left(i,j\right)\neq I_{2}\left(i,j\right),\\ 0 & \mathrm{otherwise} \end{array}\right.$$
The width and height of the two ciphered images $I_{1}$ and $I_{2}$ are denoted by *W* and *H*.

In Table 5, each gray level is coded in such a way that only eight bits are used in the coding. The obtained calculations demonstrate that the percentage of the NPCR is acceptable, as it is close to 99.6%, while the percentage of the UACI is close to 33.4% is also acceptable for the assigned images. Therefore, we can conclude that our encryption algorithm can survive against differential attack.

#### 5.2.2. Key Sensitivity Test

Some key sensitivity tests are carried out in this subsection in order to check our proposed algorithm. Figure 4a–c shows the decrypted images of the ciphered images of lena, barbara, and house using the proper values of the secret key. In the secret key, if the value of *a* is changed to 1.99999999999999 and the other values of the secret key are unchanged, the results give the decrypted images shown in Figure 4d–f, which are completely unreadable and different from the original ones. Figure 4g–i shows the decrypted images of the ciphered images with a similar security key except c=0.49999999999999. Based on those obtained results, one can conclude that any small change in one of the security keys leads to a bad image being produced, and then, the original plain image cannot be retrieved. Therefore, our algorithm possesses strong security keys.

### 5.3. The Analysis of Our Key Space

A strong encryption algorithm has a large enough key space. For high security, the key space should be greater than $2^{100}$ [2,29]. The proposed algorithm uses two initial values for $x_{0}$ and two initial values for $r_{0}$. The logistic map is utilized for row shuffling and column shuffling. In addition, there are six initial values, $x_{1,0},x_{2,0},a,b,c$, and *k*, which are used in the two-dimensional chaotic economic map. Therefore, our security keys consist of ten values. Therefore, using a precision of $10^{-14}$ [2], the size of the proposed algorithm key space is $10^{140}$, and it is more than $2^{100}$. Table 6 shows that our proposed algorithm has a key space larger than the key spaces in [1,6,21,22]. Therefore, the key space of our algorithm is large enough to resist brute-force attacks. This makes our proposed algorithm sufficiently robust to oppose a wide range of brute-force attacks.

### 5.4. Noise Attacks

The security of our algorithm against noise is tested. Two famous types of noises are incorporated into the encrypted images. The first is Gaussian noise with variances of 0.01 and 0.1. The second is salt and pepper with densities of 0.05 and 0.1. The Mean Square Error (MSE) and Peak Signal to Noise Ratio (PSNR) are used for measuring.

Assume that *X* and *Y* denote the plain image and decrypted image for the ciphered image after the noise is incorporated. MSE and PSNR are calculated by the following equations:(10)$$\mathrm{MSE}=\frac{1}{M\times N}\sum_{i=1}^{M}\sum_{j=1}^{N}{\left(X_{ij}-Y_{ij}\right)}^{2}$$and:(11)$$\mathrm{PSNR}=10\times \log_{10}\left(\frac{I^{2}}{MSE}\right)$$ where *I* is the maximum possible pixel value of an image.

The high similarity between the plain image and the decrypted image has been measured using the PSNR. Table 7 and Figure 5 show the results. The results show that the images with salt and pepper noise are better than the images with Gaussian noise because the PSNR is greater than 18.6. In addition, the histograms of the decrypted images and the plain images are approximately similar.

### 5.5. Contrast Analysis

The difference investigation of the image permits the observer to entirely understand the texture of images. The ciphered image has high contrast levels as a result of the high randomness given by the map (1) in the process of encryption. We have evaluated the contrast of the ciphered image and the effectiveness of Map (1) in image encryption applications. This can be carried out by the following formula:(12)$$C=\sum_{i,j}{\left|i-j\right|}^{2}p\left(i,j\right)$$ where p(i,j) is used to refer to the number of gray levels in the co-occurrence matrices.

The contrast analyses of both the plain and cipher images for lena (256×256), barbara (256×256), and house (256×256) are shown in Table 8. Table 9 shows that the contrast analysis of the introduced algorithm overcomes the algorithm in [6]. It has the best result.

### 5.6. Gray Value Degree Analysis

Measuring the gray difference of a pixel requires one to know its four neighbors. This can be carried out using the following formula:(13)$$G\left(x,y\right)=\frac{\sum {\left[I\left(x,y\right)-I\left(\overset{´}{x},\overset{´}{y}\right)\right]}^{2}}{4},\left(\overset{´}{x},\overset{´}{y}\right)=\left\{\begin{array}{c} \left(x-1,y\right)\\ \left(x+1,y\right)\\ \left(x,y-1\right)\\ \left(x,y+1\right) \end{array}\right.$$ where I(x,y) refers to the value of a pixel at location (x,y) and $I(\overset{´}{x}-\overset{´}{y})$ denotes its four neighboring pixels. Now, the average difference of the whole image is measured by the following:(14)$$\bar{I}\left(x,y\right)=\frac{\sum_{x=2}^{M-1}\sum_{y=2}^{N-1}G\left(x,y\right)}{\left(M-2)\times (N-2\right)}$$ where *M* and *N* refer to the row and column numbers in an image. Using (11) and (12), the Gray Value Degree (GVD) is defined as,
(15)$$\mathrm{GVD}=\frac{\overset{´}{\bar{I}}\left(x,y\right)-\bar{I}\left(x,y\right)}{\overset{´}{\bar{I}}\left(x,y\right)+\bar{I}\left(x,y\right)}$$ where the average neighborhood gray difference of an original image is denoted by $\overset{´}{\bar{I}}$, while $\bar{I}\left(x,y\right)$ refers to those of the encrypted image. Table 10 presents the degree values of the computed gray value degree for distinct images using the proposed algorithm. One can see from Table 10 that our obtained degree values are close to the ideal value, which is one. Table 11 shows some validations of the proposed algorithm versus existing algorithms in the literature.

### 5.7. Peak Signal to Noise Ratio Analysis

PSNR is used to carry out the evaluation of the encryption algorithms. It is applied to both images, the original image and the encrypted one. It considers the original image as a signal, while the encrypted image as a noise. PSNR is calculated by using Equation (Equation 11). Table 12 shows the PSNR values of some different images. It is shown that those values are low, which means it is difficult to retrieve the original image from the encrypted one in the case of hacker attacks.

### 5.8. Computational Complexity

We make a comparison of the computational complexity between the traditional DES, AES, and other algorithms in the literature including our proposed algorithm. The results appear in Table 13. Based on Table 13, we can concluded that the time of our algorithm is less than the traditional DES and AES algorithms and the other algorithms in the literature and is completely acceptable for encryption of an image.

## 6. Conclusions

The current paper introduced a new cryptographic algorithm based on a two-dimensional chaotic economic map and logical operator bit-wise XOR. The size of the key space for our proposed algorithm is $10^{140}$, which allows our algorithm to survive against attacks. The numerical experiments have shown that the algorithm is very sensitive to the security keys, and hence, any bit change or random guessing about the proper value of the security keys fails to retrieve the original image. The obtained results by the proposed algorithm have shown that the ciphered image is random-like since the information entropy of the cipher image of size 256×256 is very close to the theoretical value of eight, the NPCR is close to 99.6%, and UACI is close to 33.4%. Therefore, our algorithm is extremely sensitive to any small change in the original image. According to the obtained results, we may conclude that the proposed algorithm owns a high level of security that allows it to prevent any attempts of attack by hackers.

## Figures and Tables

**Figure 1 entropy-21-00044-f001:**
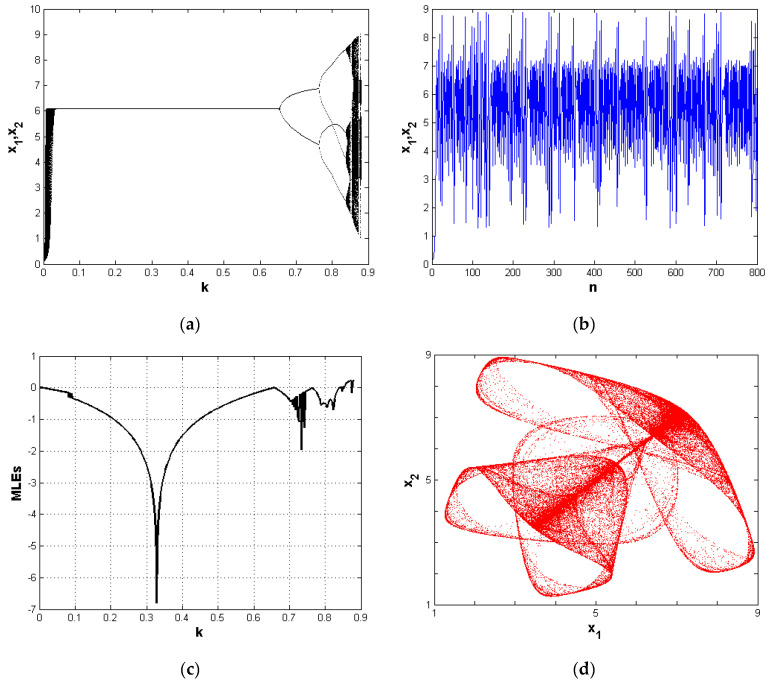
(**a**) Bifurcation diagram with respect to the parameter *k* at $\left(x_{1,0},x_{2,0},a,b,c\right)=\left(0.11,0.12,2,0.5,0.5\right)$; (**b**) time series for the system’s variables at $\left(x_{1,0},x_{2,0},k,a,b,c\right)=\left(0.11,0.12,0.87,2,0.5,0.5\right)$; (**c**) the Lyapunov exponent with respect to the parameter *k* at $\left(x_{1,0},x_{2,0},a,b,c\right)=\left(0.11,0.12,2,0.5,0.5\right)$; (**d**) chaotic behavior of the system at the parameters $\left(x_{1,0},x_{2,0},k,a,b,c\right)=\left(0.11,0.12,0.87,2,0.5,0.5\right)$.

**Figure 2 entropy-21-00044-f002:**
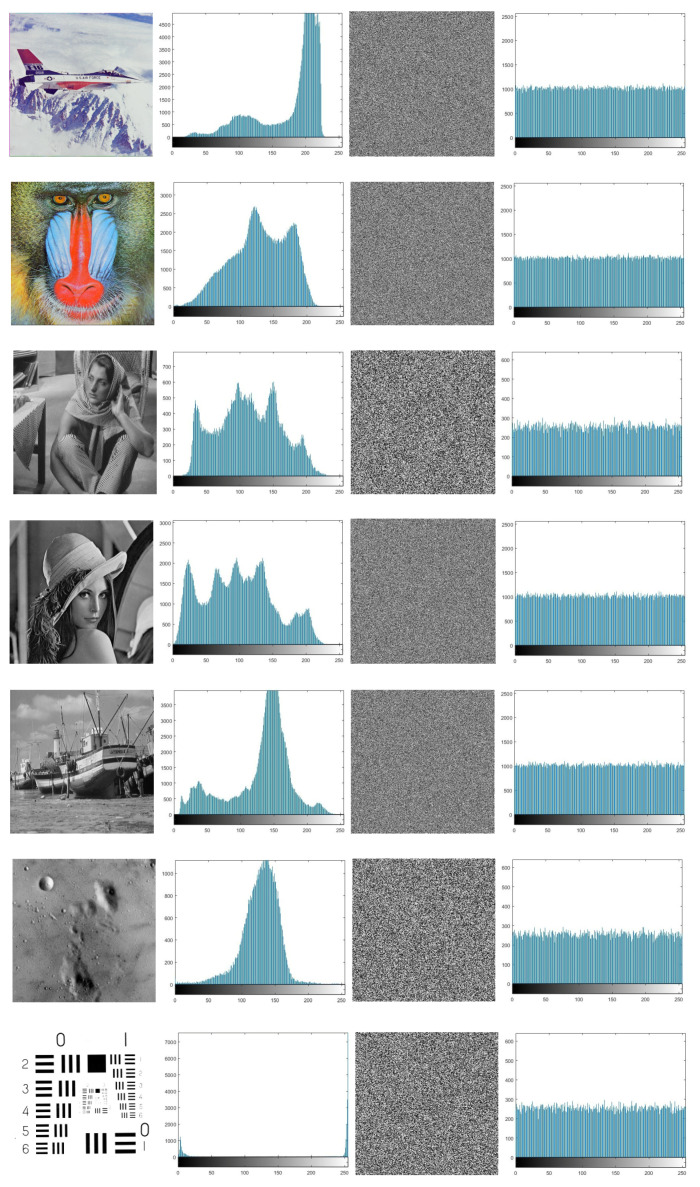
Histogram diagram for plain and cipher images at $a=2,b=0.5,c=0.5,x_{1,0}=0.11,x_{2,0}=0.12$, and k=0.87.

**Figure 3 entropy-21-00044-f003:**
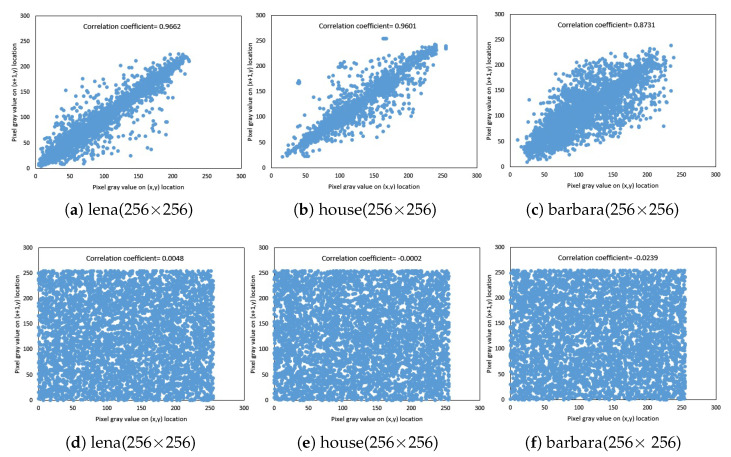
(**a**–**c**) Correlation analysis of plain images and (**d**–**f**) correlation analysis of cipher images at $a=2,b=0.5,c=0.5,x_{1,0}=0.11,x_{2,0}=0.12$, and k=0.87.

**Figure 4 entropy-21-00044-f004:**
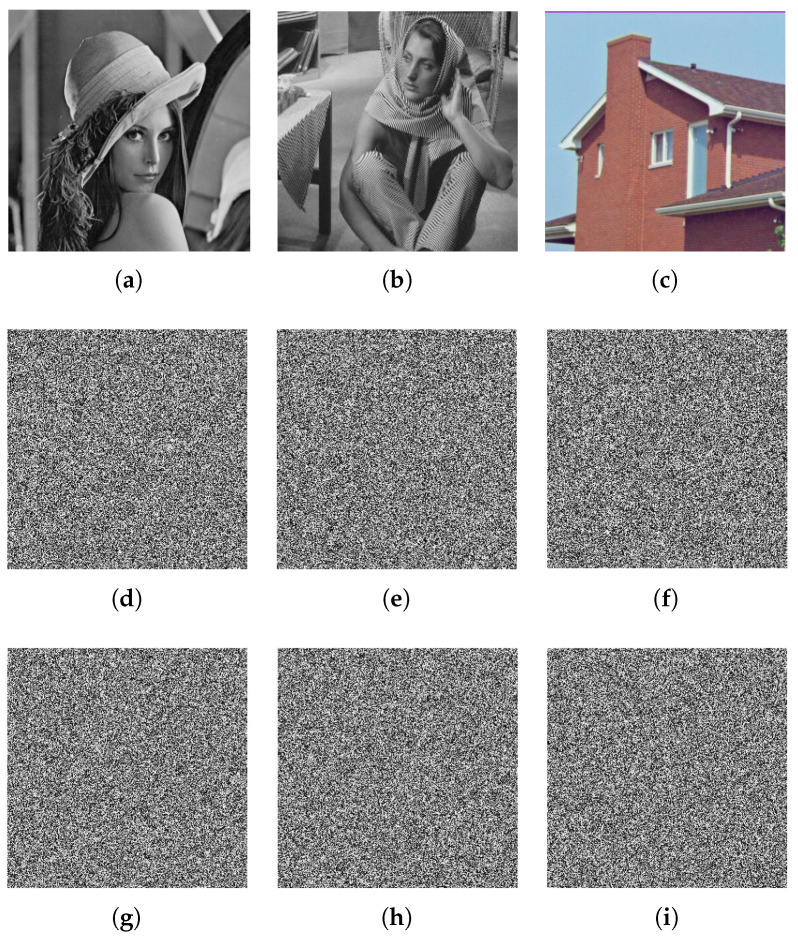
Decrypted images of lena, barbara, and house for the proper secret key (**a**–**c**), (**d**–**f**) decrypted images at $a=1.99999999999999,b=0.5,c=0.5,x_{1}\left(0\right)=0.11,x_{2}\left(0\right)=0.12$, and k=0.87, and (**g**–**i**) decrypted images at $a=2,b=0.5,c=0.49999999999999,x_{1,0}=0.11,x_{2,0}=0.12$, and k=0.87.

**Figure 5 entropy-21-00044-f005:**
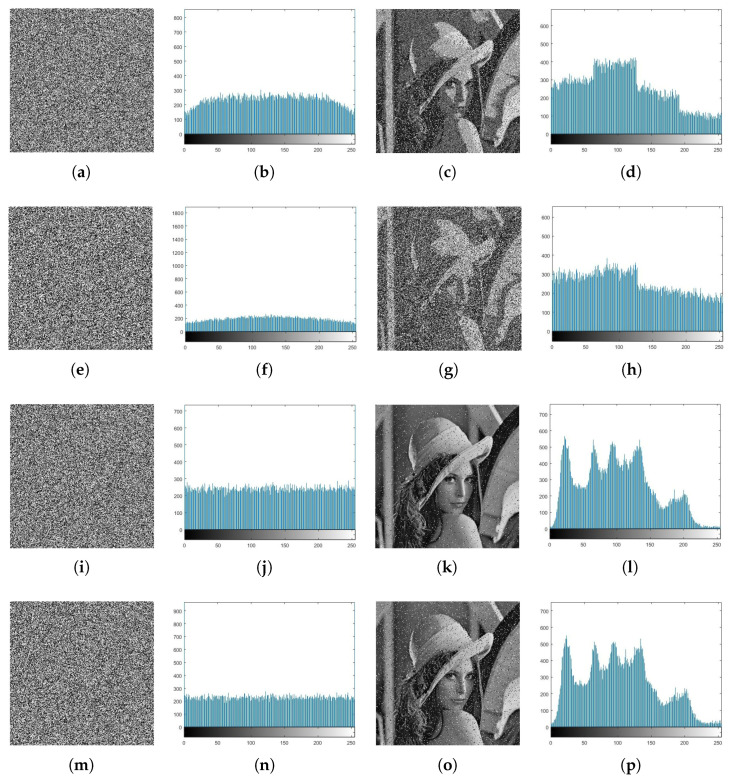
(**a**,**e**,**i**,**m**) Encrypted images after adding Gaussian and salt and pepper, (**b**,**f**,**j**,**n**) corresponding histograms of the encrypted images, (**c**,**g**,**k**,**o**) decrypted images at $a=2,b=0.5,c=0.5,x_{1,0}=0.11,x_{2,0}=0.12$, and k=0.87, and (**d**,**h**,**l**,**p**) corresponding histograms of decrypted images.

**Table 1 entropy-21-00044-t001:** $\chi^{2}$ values of plain and cipher images at $a=2,b=0.5,c=0.5,x_{1,0}=0.11,x_{2,0}=0.12$, and k=0.87.

Image	$\chi^{2}$	
	Plain Image	Cipher Image
Lena	$4.0659\times 10^{4}$	255.4371
Cameraman	$1.1097\times 10^{5}$	255.9152
Barbara	$3.5245\times 10^{4}$	258.9021

**Table 2 entropy-21-00044-t002:** Information entropy tests for cipher images at $a=2,b=0.5,c=0.5,x_{1,0}=0.11,x_{2,0}=0.12$, and k=0.87.

Image			Theoretical Block Entropy	
	**Global**	**Actual Block**	α=0.01	α=0.05
	**Entropy**	**Entropy**	7.16276745	7.16634107
lena (256 × 256)	7.9981	7.1921	Pass	Pass
lena (512 × 512)	7.9994	7.1862	Pass	Pass
barbara (256 × 256)	7.9973	7.1801	Pass	Pass
airplane (512 × 512)	7.9996	7.1817	Pass	Pass
boat (512 × 512)	7.9995	7.1849	Pass	Pass
house (256 × 256)	7.9984	7.1829	Pass	Pass
baboon (512 × 512)	7.9995	7.1827	Pass	Pass
moon surface (256 × 256)	7.9982	7.1819	Pass	Pass
resolution chart (256 × 256)	7.9980	7.1812	Pass	Pass

**Table 3 entropy-21-00044-t003:** Correlation coefficients of two adjacent pixels of the plain image and cipher image at $a=2,b=0.5,c=0.5,x_{1,0}=0.11,x_{2,0}=0.12$, and k=0.87.

Image		Plain Image	Cipher Image
lena (128 × 128)	H	0.8906	−0.0102
	V	0.9518	−0.0230
	D	0.8512	0.0119
lena (256 × 256)	H	0.9351	0.0005
	V	0.9692	0.0017
	D	0.9181	−0.0025
lena (512 × 512)	H	0.9659	−0.0091
	V	0.9837	−0.0198
	D	0.9552	−0.0062
barbara (256 × 256)	H	0.8105	−0.0088
	V	0.8797	−0.0179
	D	0.8335	−0.0054
airplane (512 × 512)	H	0.9566	0.0096
	V	0.9600	0.0053
	D	0.9237	0.0063
boat (512 × 512)	H	0.9385	−0.0025
	V	0.9669	0.0186
	D	0.9225	−0.0111
house (256 × 256)	H	0.9812	0.0003
	V	0.9660	−0.0178
	D	0.9402	0.0050
baboon (512 × 512)	H	0.9121	0.0110
	V	0.8634	0.0019
	D	0.8282	0.0118
moon surface (256 × 256)	H	0.8859	−0.0026
	V	0.9316	0.0100
	D	0.8988	−0.0029
resolution chart (256 × 256)	H	0.8828	0.0007
	V	0.8793	0.0106
	D	0.7524	0.0159

**Table 4 entropy-21-00044-t004:** Correlation coefficients of two adjacent pixels of the cipher image for the proposed algorithm at $a=2,b=0.5,c=0.5,x_{1,0}=0.11,x_{2,0}=0.12$, and k=0.87 and other algorithms.

Image		Cipher Image						
		Proposed	[6]			[1]	[21]	[22]
			Case I	Case II	Case III			
	H	0.0005	0.0122	0.0075	−0.0038	0.0077	-	−0.0230
lena	V	0.0017	−0.0456	−0.0079	0.0093	0.0168	-	0.0019
(256×256)	D	−0.0025	−0.0188	−0.0093	−0.0189	0.0104	-	−0.0034

**Table 5 entropy-21-00044-t005:** Comparison of the Number of Pixels Change Rate (NPCR) and Unified Average Changing Intensity (UACI) of the lena, barbara, and house images with size 256×256 using our proposed algorithm and other algorithms.

Cipher Algorithm	NPCR	UACI
Theoretical expected value	99.61%	33.46%
Proposed (lena)	99.60%	33.43%
Proposed (barbara)	99.62%	33.45%
Proposed (house)	99.60%	33.40%
[1] (lena)	99.59%	30.63%
[21] (lena)	99.60%	33.46%
[22] (lena)	99.62%	33.51%

**Table 6 entropy-21-00044-t006:** Key space comparison.

Algorithm	Ours	[1]	[6]	[21]	[22]
Key Space	$10^{140}$	$10^{140}$	$10^{84}$	$2^{256}\approx 1.16\times 10^{77}$	$2^{210}\approx 1.65\times 10^{63}$

**Table 7 entropy-21-00044-t007:** Noise attack results.

Noise	MSE	MSE	PSNR	PSNR
	(Our Algorithm)	[30]	(Our Algorithm)	[30]
Gaussian noise with variance = 0.01 and mean = 0	2321.4	4410.1	14.5	11.7
Gaussian noise with variance = 0.1 and mean = 0	5201.2	5631.4	11.0	10.6
Salt and pepper noise with density 0.05	437.9	869.9	21.7	18.7
Salt and pepper noise with density 0.1	893.1	1829.6	18.6	15.5

**Table 8 entropy-21-00044-t008:** Contrast analysis of plain and cipher images at $a=2,b=0.5,c=0.5,x_{1,0}=0.11,x_{2,0}=0.12$, and k=0.87.

Image	Contrast	
	Plain Image	Cipher Image
lena (256×256)	0.3563	10.6201
house (256×256)	0.1741	10.5431
barbara (256×256)	0.9463	10.6986

**Table 9 entropy-21-00044-t009:** Contrast analysis of the plain (lena (256×256)) and cipher image using the proposed algorithm at $a=2,b=0.5,c=0.5,x_{1,0}=0.11,x_{2,0}=0.12$, and k=0.87 and other algorithms.

Image	Contrast							
	Cipher Image							
	Plain	Proposed	[6]			[21]	[22]	[1]
	Image	Algorithm	Case I	Case II	Case III			
lena	0.3563	10.6201	10.1655	10.3909	10.3971	10.5034	10.4723	10.4767

**Table 10 entropy-21-00044-t010:** Gray Value Degree (GVD for different test images.

Image	GVD Value
lena	0.9653
barbara	0.9571
house	0.9755
moon surface	0.9756
resolution chart	0.9482

**Table 11 entropy-21-00044-t011:** GDV analysis of the proposed algorithm with other algorithms.

Image	GVD Value			
	Proposed Algorithm	Arnold’s	[31]	[32]
lena	0.9653	0.8900	0.9540	0.9624

**Table 12 entropy-21-00044-t012:** PSNR values for different test images.

Image	PSNR Value
lena	8.5557
barbara	9.1158
house	9.2541
moon surface	10.1982
resolution chart	4.9300

**Table 13 entropy-21-00044-t013:** The speed of encryption for each algorithm. DES, Data Encryption Standard.

Algorithm	Encryption Time (in Seconds)
Proposed Algorithm	0.1309
DES	0.6305
AES	0.2173
[21]	0.1493
[22]	0.2021
[1]	0.5463

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
