# Peer review of "An Algorithm of Image Encryption Using Logistic and Two-Dimensional Chaotic Economic Maps"

_entropy, 2019, doi:10.3390/e21010044_

Round 1

Reviewer 1 Report

The paper intends to propose a new cryptographic algorithm for image encryption based on two chaotic systems. In my view the paper has several flaws as follows:

1.       The paper is hard to read, both in terms of grammar and structure of the text. Some examples (only from abstract): a) the sentence started in line 2 is confusing; b) in line 5 it is written “robust” instead of “robustness”; c) in line 8 it is written “leaded” instead of “led”; d) in line 9 it is written “has characterized” instead of “has been characterized”; etc.

2.       The proposed algorithm is very close to the one presented by the same authors in [20] “Cryptographic algorithm based on pixel shuffling and dynamical chaotic economic map”. The differences must be detailed.

3.       I have doubts that the so called Chaotic Economic Map (CAM) is indeed chaotic. I found no reference with a proof for this. If the authors want to use this nonlinear system they have to prove it is indeed chaotic.

4.       For which parameter values system (1) is chaotic? See what happens for k=0 for example, or for a=c and b=0.

5.       In Section 5.3 there are some mistakes: a) why (a-c) is not considered to be a single parameter in Eq. (1) (let’s say d)? b) Why in subsection 5.3 n is considered as a parameter (part of the secret key)?

6.       An encryption function must be bijective (invertible). Some of the steps in the encryption algorithm seem to lead to a non-bijective function (see step 1 and step 2). Moreover the decryption algorithm is not proved to be the inverse of the encryption algorithm. In these circumstances (if the encryption function is not invertible) all the evaluation studies presented in the paper are irrelevant.

7.       Some notations are not explained. See for example equations (5), (6) or (7).

8.       The empirical results presented in subsection 5.8 cannot sustain the idea that the proposed algorithm is better from the computational point of view.

9.       The sentence started in line 237 is strange and unproved. There are cryptographic attacks that can be done even for a large key space.

10.   The security of the proposed algorithm is uncertain. What types of security attacks were considered (see the line 11 in the abstract)?

11.   Line 23: bifurcation is not a feature of chaotic system. It can be used as a method in proving that a system is chaotic, but it is not a feature.

12.    The “Authors Contributions” section is not presented at the end of the paper as requested by journal’s Information for Authors.

Author Response

My response is attached as a pdf File

Reviewer 2 Report

The authors are proposing an original image encryption algorithm based on a two-dimensional chaotic economic map. The theoretical and experimental results are showing an improvement of the state-of-the-art in key characteristics of an image encryption algorithm.

The paper needs several improvements in what concerns the English language. Some examples are provided here only for the abstract: "The robust" - robust is an adjective not a noon; "are analyzed entirely" very strong statement - only partially analyzed (e.g. time consumption is not discussed in details); "the introduced algorithm has characterized" is instead has; "resists different" to different.   A random picked text from inside: "A strong encryption algorithm has a large enough key space" - large enough for what? compared with what? unclear.

The tables are containing a lot of numbers and the texts associated to them are insufficiently explaining what is good/bad both for the proposed as well as referred existing algorithms Moreover, it is unclear in Table 5 what is "Expected value".

Please revise (11). It is unclear that the sum is probably done over all neighbours and the use of minus to explain the variants is not adequate.

Author Response

My response is attached as a pdf file .

Reviewer 3 Report

The manuscript describes an image encryption system based on chaotic systems. Some comments/suggestions follow:   

- Section 1:
      * Phrase "Encryption schemes have been introduced based on chaotic models to handle that issue and to provide such secure routes" requires a reference illustrating it.
      * "The data encryption algorithm is a traditional scheme and has been used for encrypting images". What is "the data encryption algorithm"?
      * "It has been recently adopted by researchers in the process of encryption." -> This affirmation requires references.
   - Section 3:  
      * Typo -> "will be change" should be "will be changed".
      * In this algorithm, what is the secret key ? It is not clearly defined, and how to create a secure secret key is not described.

   - Section 5.2.2.
     * ".. using the proper values of the secret key" -> What are these "proper values"?. They have not defined.  
     * It seems that "a" parameter is part of the secret key. This value is a floating point value, and it is not a good idea to use floating point values for secret keys (representation and rounding errors can avoid to decrypt the image although the receiver knows the correct key):
   - Section 5.3.  
     * In this section, composition of the secret key is outlined, but, again it is not clearly stated.  
     * Precision of parameters, which are part of the secret key is fixed to 10**-14. Why have you used this value? This should be clarified in the text.  

-    The article contains several typos and grammatical errors. It should be thoroughly revised.

Author Response

My response is attached as a pdf file .

Round 2

Reviewer 1 Report

The authors have successfully addressed all my previous comments and concerns.

Author Response

submitted before 

Reviewer 2 Report

The authors are proposing an original image encryption algorithm based on a two-dimensional chaotic economic map. The theoretical and experimental results are showing an improvement of the state-of-the-art in key characteristics of an image encryption algorithm.

The paper content was improved after the review. No further concerns are generated by the new version.

Author Response

submitted before in round 1

Reviewer 3 Report

Comments and suggestions follow:

    - Section 1. The phrase " The data encryption algorithm is a traditional ... " remains uncorrected. If yout are referring to the "Data Encryption Standard (DES) Algorithm", please, write it correctly.

    - Section 3.1.

          * Now, I undestand how the secret key is derived. Nevertheless, I am not sure if it should be described in more detail, describing step by step how each or the ten parameters is computed. 

          * What range of values can be used for the parameters involved in the secret key? All possible values obtained from expressions (2) and (3) are valid ?.

         * Line 129. Typo:  "u=1,2,...,10, receptively" should be "u=1,2,...10, respectively".

- Although "Most research papers in this area used 10^-14 as a double-precision for
numbers to be accurate during the computations." , a paragraph justifying it or including a reference will be interesting for the reader.

- There are typos and grammatical errors in the text yet.
